# Clinical Results of the MINIject Implant for Suprachoroidal Drainage

**DOI:** 10.3390/jcm13102831

**Published:** 2024-05-11

**Authors:** Timothy Gläser, Daniel Böhringer, Charlotte Evers, Philip Keye, Thomas Reinhard, Jan Lübke

**Affiliations:** Eye Center, University Hospital Freiburg, 79106 Freiburg im Breisgau, Germanythomas.reinhard@uniklinik-freiburg.de (T.R.); jan.luebke@uniklinik-freiburg.de (J.L.)

**Keywords:** glaucoma, MIGS, suprachoroidal drainage

## Abstract

**Objective:** This retrospective study evaluated the safety and efficacy of the new minimally invasive MINIject implant placed in the suprachoroidal space. The aim was to assess its impact on intraocular pressure (IOP) reduction and complication rate. **Methods:** 18 eyes from 18 patients with insufficiently controlled glaucoma received the implant using topical medications. Outcomes were changes in IOP, change in IOP medication, need for other glaucoma surgery, and rate of adverse events. **Results:** IOP reduced by 15% (*p* < 0.05) following MINIject implantation. IOP medication decreased from 3 to 1 agent (*p* < 0.05). Four patients (22%) required other glaucoma surgery while we did not observe any clinically relevant adverse event. **Conclusions:** This retrospective study indicates that MINIject implants may be a safe and effective means of reducing IOP together with a reduction in IOP medications in most patients. Larger prospective studies with longer follow-ups are necessary to confirm our results, though.

## 1. Introduction

Glaucoma remains a leading cause of blindness globally. An estimated 80 million people are affected worldwide [1]. While its pathophysiology is multifactorial, elevated intraocular pressure (IOP) remains the primary risk factor for its onset and progression. The two types of glaucoma included in this study are primary open-angle glaucoma (POAG) and pseudoexfoliative glaucoma (PXG). POAG is the most common glaucoma variant and presents with anatomically normal ocular structures but increased outflow resistance. PXG is a type of secondary glaucoma where the deposition of fibrillary material in the trabecular meshwork leads to increased outflow resistance. Current treatment options involve medical or surgical intervention with the aim of lowering IOP. This is effectuated by either a reduction of aqueous humor production or increasing its outflow out of the anterior chamber. Different treatment approaches vary in both their efficacy and possible side effects.

Apart from shunting procedures from the anterior chamber to the subconjunctival or sub-tenon space, glaucoma surgery targets one of two physiological outflow pathways. The conventional outflow pathway consists of the trabecular meshwork and Schlemm’s canal. The unconventional, or uveoscleral, pathway leads to the suprachoroidal space (SCS). A third, uveolymphatic pathway is still poorly understood and not utilized clinically [2]. The SCS is a potential space bound by the sclera on the outside, the choroid on the inside, and the loosely connected ciliary body band anteriorly [3]. Being a potential space, it normally occludes between the sclera and choroid. Its location further from the ocular surface allows for better cosmetic results as well as a reduced risk of endophthalmitis compared to bleb-forming surgeries [4].

The MINIject implant is a novel type of implant used in minimally invasive glaucoma surgery (MIGS). The implant is made of a proprietary silicone-based material, which is meant to have a high degree of biocompatibility [5]. Promising results with a meaningful lowering of IOP and low complication rates were shown in the first clinical trials [6]. These trials were carried out by the device manufacturer (iSTAR Medical, Wavre, Belgium). Long-term results remain to be published, with the longest follow-up period published at 2 years [7,8].

This retrospective study provides the first independent clinical data on the real-world usage of the MINIject implant at the Freiburg University Eye Hospital.

## 2. Materials and Methods

This retrospective analysis encompasses all MINIject implant procedures conducted at Freiburg University Eye Hospital throughout the year 2022. To minimize bias, only the first eye was included in cases where surgery on both eyes were performed. All procedures were performed as inpatient cases by a single, seasoned glaucoma consultant.

All patients included in this study were seen initially in the glaucoma clinic. Past medical history, current medication, best-corrected visual acuity, IOP, and a full anterior and posterior segment exam were completed. Further diagnostics, including perimetry and optical coherence tomography, were obtained on a case-by-case basis.

16 of the patients’ eyes had automated static computer perimetry conducted on an Octopus 300 perimeter (Haag-Streit, Köniz, Switzerland). At the consultant’s discretion, 2 patients had manual dynamic perimetry conducted on a Goldmann perimeter (also by Haag-Streit). Brusini’s Enhanced Glaucoma Staging System (GSS2) was used to quantify glaucomatous damage alongside the mean deviation (MD) [9].

Endothelial cell density (ECD) was only measured in case of any clinical sign of an endothelial pathology. Patients with endothelial pathology in their past medical history were not considered for MINIject implantation.

IOP was measured using a Goldmann application tonometer (GAT) before the application of mydriatics. A comprehensive evaluation of eligibility criteria for the MINIject implant was assessed. The risks and alternatives associated with the procedure were discussed with the patients before informed consent was obtained prior to surgery. Patients were informed of the novel nature of the implant.

Patients were admitted to the hospital one day before surgery. As is standard practice at the university eye hospital in question, an all-day IOP profile, including supine measurements at night, was obtained. These measurements were conducted roughly every 4 h starting on the preoperative day. The 12:00, 16:00, and 20:00 measurements were obtained using GAT, and the 24:00 and 07:00 measurements were conducted in a supine position using a handheld rebound tonometer. The longer interval between the supine measurements is chosen to allow the patients better rest during the night.

Postoperative care consisted of an ofloxacin eyepatch immediately following surgery with 1–2 days of convalescence in the university hospital ward. Regular post-operative check-ups were performed during this time including visual acuity testing, IOP measurements and assessment of postoperative inflammation as well as complications. Barring contraindications, topical steroid eye drops were administered five times daily from the first postoperative day forward and were tapered off weekly. Follow-up examinations were scheduled at the consultant’s discretion every 1–2 weeks until satisfactory findings permitted referral to a local practitioner.

Data were entered from the patient’s records into a relational research database maintained by the University Eye Center. Further processing was conducted in an anonymized form with data exported from the research database.

Statistical analysis was performed using the open source software “R” version 4.1.1 by the R Foundation for Statistical Computing. The alpha level was set to 0.05. Normality was tested with the Shapiro-Wilk test. In the case of normally distributed data, the paired T-test was used, and for non-normally distributed data, the Wilcoxon test was employed. A Kaplan Meier analysis was conducted with event-free survival, which was defined as no need for further drainage surgery.

Preoperative IOP was defined as average of the six IOP measurements made from the time of indication of the procedure up to the procedure. This includes a single measurement in the glaucoma clinic and an IOP profile with 5 measurements on the day before the operation as described above. This was conducted in order to have a more representative value for the long-term average IOP value as a single measurement has limited meaningfulness.

Postoperative IOP was defined as the last measurement in the follow up visits or the last measurement before a follow-up procedure other than repositioning was necessary. No further all-day IOP profiles were obtained. The last measurement was chosen under the assumption that the expected treatment target was reached or the pressure was not expected to lower further if no more measurements were made.

Cox proportional hazard regression analysis was performed to identify protective and hazardous factors for the need for further drainage surgery.

## 3. Results

A total of 18 eyes of 18 patients were included. Patient age was 80.0 years on average, with a range from 67 to 91 years and a standard deviation of 6.7 years. All eyes were pseudophakic. A total of 14 of 18 eyes belonged to female patients. A total of 11 eyes had primary open-angle glaucoma, with the remaining 7 eyes having pseudoexfoliative glaucoma. The mean deviation of the 10 eyes that had automated computer perimetry conducted preoperatively showed a mean MD of −7.75 dB and a standard deviation of 6.72 dB. This corresponds to mean stage 2.75 in the enhanced glaucoma staging system [9].

Pre-operatively, patients were on 2.85 medications on average, with six patients having a quadruple topical therapy of beta-blockers, alpha-agonists, carbonic anhydrase inhibitors, and prostaglandin-analogues (Table 1).

The mean IOP at the time of indication of the procedure across all patients was 23.8 mmHg. A total of 14 out of 18 patients did not require further surgery. In this group, the preoperative pressure was 21.2 mmHg, and the postoperative pressure was 18.0 mmHg, representing a decrease of 15.1% (*p* < 0.05, Table 2). The mean number of topical medications dropped from 2.7 medications to 1.1 medications (*p* < 0.05). The mean follow-up was 107.9 days, with a range of 35 to 242 days.

4 out of 18 eyes required further drainage surgery (3 XEN-Implants, 1 Paul-Implant) due to insufficient pressure regulation (Figure 1). Kaplan Meier survival analysis predicts a survival rate of 94% (95%CI: 84–100%) to remain free of such interventions after the mean follow-up period of 107.9 days (Figure 2).

No cases of clinically significant hypotony, macular edema or hemorrhage were observed. A total of 5 eyes required repositioning of the implant due to inadequate suprachoroidal drainage and elevated pressure. 2 of those 5 went on to receive drainage implants while the other 3 eyes required no further surgery.

Cox proportional hazards regression analysis showed a protective effect of POAG relative to PEX glaucoma with a hazard ratio of 0.44 (95% CI 0.04–4.97) for further drainage surgery. Previous surgery via trabectome was associated with a higher risk of further surgery with a HR of 1.34 (95% CI 0.12–15.07).

## 4. Discussion

The mean IOP decrease was 3.2 mmHg with 1.6 fewer topical medications. A total of 14 out of 18 eyes, or 78%, achieved a satisfactory IOP and did not require further drainage surgery.

The STAR-studies reported a mean decrease in IOP of 9 mmHg while our study showed a decrease of 3.2 mmHg, or 15%, of baseline. Better efficacy was seen in the reduction of topical medication by 59% in eyes which did not require further surgery. The actual pressure reducing potential of the implant may be masked by the reduction in IOP lowering eye drops.

No cases of clinically significant hypotony were reported. While not uncommon in glaucoma surgery and often resolving spontaneously, hypotony can prove difficult to manage in some cases. The micropore structure of the implant and the finite volume of the suprachoroidal space may act to limit the potential for large pressure drops. Conversely, this fact also limits the therapeutic potential for lowering IOP to a desired range.

Similarly, no other common complications from intraocular surgery, such as hemorrhage or clinically significant macular edema, were observed. The device was removed from one eye during a follow-up procedure (XEN implantation) as it was found to have partially dislocated into the anterior chamber.

Due to the low number of eyes, only limited conclusions can be drawn from the complication rates.

Pseudoexfoliation leads to a slow buildup of material in the trabecular meshwork, thereby increasing outflow resistance and intraocular pressure. Pseudoexfoliative material is thought to obstruct other glaucoma implants such as the XEN (AbbVie, North Chicage, IL, USA) or Preserflo (Glaukos, Aliso Viejo, CA, United States) implant (inner lumen of 45/63 µm and 70 µm). A similar mechanism is conceivable in the micropore structure of the implant and might manifest itself at longer follow-up intervals. This may have manifested itself in the lower hazard ratio for further drainage surgery in POAG patients. With pseudoexfoliative glaucoma being the most common secondary open-angle glaucoma, this may represent another important risk factor when choosing patients for MINIject implantation.

Compared to other micro-invasive surgical procedures that target the trabecular meshwork and Schlemm’s canal we saw a smaller decrease in IOP compared to reported 12 month follow-up data in a systematic meta-analysis [10]. Similarly, the XEN implant, Preserflo, and the classic trabeculectomy led to a larger decrease in IOP after 12 months [11,12]. Table 3 summarizes these results compared to our own data.

The Solx Gold Micro Shunt (SOLX, Waltham, MA, USA), another supraciliary/suprachoroidal implant, showed promising early results while having a low long-term survival rate [13,17]. Membrane formation, which obstructed the drainage pathway out of the anterior chamber, was described to occur in a majority of failure cases [18].

Two related devices, the Esnoper V2000 and Esnoper Clip (AJL Ophthalmic S.A., Araba, Spain) are implants used to create a shunt from the anterior chamber into the suprachoroidal space via a deep sclerotomy. While more invasive than the MINIject procedure, these implants utilize the same outflow pathway. Similarly to the Solx Gold Micro Shunt, initial 1-year follow-up data is promising, though long-term data are not yet available [14,15].

Little data is available on the iStent Supra (Glaukos, Aliso Viejo, CA, USA), a small shunt used for implantation ab interno. The single prospective study available showed favorable outcomes with a good safety profile [16]. Due to the three-fold approach with implantation of both the regular trabecular iStent as well as the iStent Supra and topical prostaglandin eye drops, the effect of the suprachoroidal implant alone is difficult to assess.

The latest supraciliary device with FDA approval was the CyPass micro-stent (Alcon, Geneva, Switzerland). The 5-year data showed an IOP decrease of 8.4 mmHg. However, this was achieved through a combined procedure, including cataract surgery. The control group, which received only cataract surgery, showed a decrease of 8.0 mmHg but required more topical medications (44.0% compared to 28.3% medication-free at 5 years). The CyPass implant was withdrawn from the market after the 5-year data in the COMPASS XT study showed subclinical but significant and progressive endothelial cell loss after CyPass implantation [19,20].

With cataract surgery alone lowering IOP by approximately 5.3 to 8.5 mmHg according to a recent review, these data suggest a similar IOP-lowering potential between CyPass and MINIject owing to their common mode of operation [21]. Endothelial cell density was not measured routinely during the clinical visits and therefore could not be assessed in this study. This represents a worthwhile avenue in future studies of any glaucoma implants that rest in the anterior chamber and could potentially have contact with the corneal endothelium. This will be of particular importance in future studies with longer follow-up periods where a different set of complications may come to the forefront irrespective of an implants efficacy.

Other glaucoma implants are probably also associated with at least some endothelial cell loss. Data from XEN implants seem to suggest endothelial cell loss comparable to phacoemulsification and posterior chamber IOL implantation [22,23]. In a post hoc analysis of 5-year data for Hydrus, CyPass, and iStent inject procedures compared to phakoemulsification alone, it could be shown that the iStent had the lowest impact on ECD with the Hydrus and CyPass implants showing higher (and, in case of CyPass, accelerating) ECD loss [24]. The Preserflo MicroShunt implant similarly showed some ECD loss in most studies, with the position of the inflow end in relation to the corneal endothelium being of particular importance [25,26].

The follow-up period in this study is quite short, while long-term pressure is the most important endpoint in assessing the IOP-lowering potential of a new glaucoma procedure. The initial postoperative phase is often accompanied by transient pressure increases due to steroid response or residual blood in the trabecular meshwork, which may act to increase IOP. As all patients who received the implant in 2022 were included in this study regardless of time since the operation, these early postoperative IOP values are included and might skew the value compared to longer follow-up periods.

Future studies might compare the long-term outcomes of MINIject implants which needed repositioning compared to uncomplicated insertions. This may further inform implantation technique if significant differences do exist in these groups.

With MINIject being a new procedure, it has an associated learning curve in surgical technique and patient selection which cannot be ignored. This effect if offset by the fact that the single surgeon who performed the implants is well versed in various kinds of glaucoma surgery and possesses a high degree of proficiency.

In conclusion, the MINIject implant represents a new micro-invasive procedure that is able to lower intraocular pressure and reduce pressure-lowering medication without severe complications. In our data, most of the patients who were treated with the MINIject implant did not need additional glaucoma surgery. This positions the MINIject implant as a hopeful candidate for the currently underutilized suprachoroidal drainage pathway. It could represent a worthwhile avenue as a supplementary pathway in cases where the commonly used procedures lead to unsatisfactory IOP control.

One of the major limitations of this study is the retrospective design with a very short follow-up. Nonetheless, we think that the information regarding the need for further glaucoma surgery is valuable, as even within this short period after MINIject implantation, a relevant number of patients had to undergo surgery again. Because of the retrospective nature, not all patients could be followed up and were therefore lost for further analysis. We would assume that this biases the results but would expect patients with insufficient IOP control to present to our hospital for further treatment. Due to the short follow-up, functional data such as visual fields could not be obtained from all patients. We are aiming to achieve functional stability with all IOP-lowering surgeries, and this needs to be addressed in further studies regarding the MINIject implant. Endothelial cell density was only obtained in a few patients. In a prospective setting, this would have been an important endpoint; the study’s retrospective design and the lack of clinical routine data on endothelial cell density in our clinic were the reasons for this.

It is important to know these limitations to assess the data we presented in our study and identify room for improvements for further studies.

## Figures and Tables

**Figure 1 jcm-13-02831-f001:**
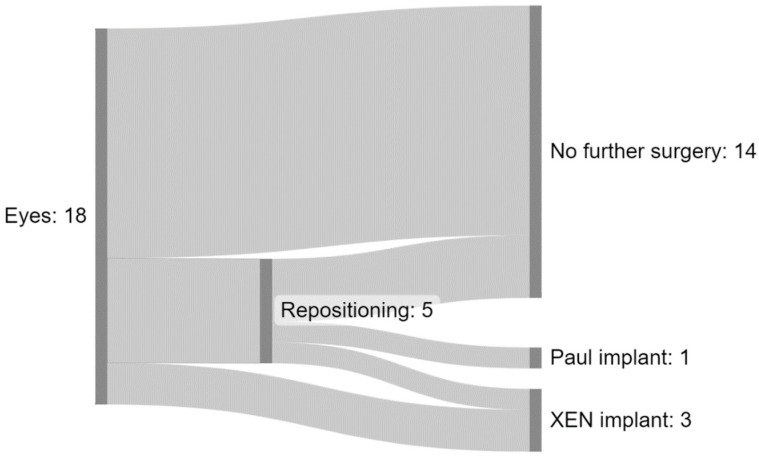
Sankey diagram showing flows of further surgical interventions. A total of 5 of 14 eyes required repositioning, of these 1 went on to receive a Paul implant and 1 a XEN implant. 2 Eyes received a XEN implant without prior repositioning.

**Figure 2 jcm-13-02831-f002:**
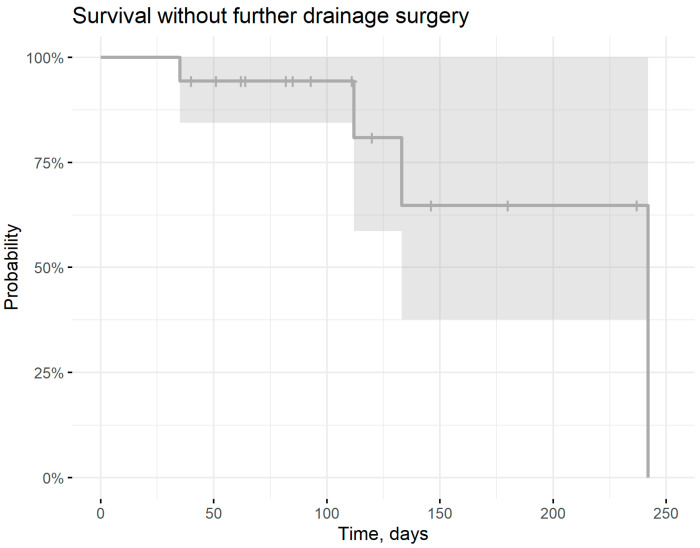
Survival free of further drainage surgery.

**Table 1 jcm-13-02831-t001:** Summary of descriptive statistics.

	Mean	Range	SD
Age, years	80.0	67–91	6.7
Pre-operative IOP, mmHg	23.8	15–36	6.1
Medications, N	2.85	0–4	1.13
Follow-up, days	107.9	35–242	61.0
Visual field mean deviation, dB	−7.75	+0.8–−18.7	6.72
GSS2 (Brusini)	2.75	1–5	1.09
Sex	Female	Male	Total
	14	4	18
Glaucoma type	POAG	PXF	Total
	11	7	18

**Table 2 jcm-13-02831-t002:** Summary eyes without further drainage surgery.

	Mean	Range	SD
Pre-operative IOP, mmHg	21.2	15–29	4.1
Post-operative IOP, mmHg	18.0	11–22	3.4
Pre-operative medications, N	2.7	0–4	1.2
Pre-operative medications, N	1.1	0–4	1.2

**Table 3 jcm-13-02831-t003:** Comparison to other interventions.

Intervention	IOP-Reduction [mmHg]
MINIject, our study	3.2
Single iStent [10]	5.4
Dual iStent [10]	5.3–11.8
CyPass + Phako [10]	8.1
Hydrus [10]	6.6
Trabectome [10]	4.6–9.7
XEN [11]	10.1–10.4
Preserflo [12]	6.8
Trebeculectomy [12]	10.0
Solx Gold [13]	9.0
Esnoper V 2000 [14]	8.1
Esnoper Clip [15]	9.0
2× iStent, 1× iStent Supra, Prostaglandin [16]	8.3

## Data Availability

The datasets presented in this article are not readily available they represent patient medical records and as such are not made available outside of the research institution.

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
