# Peer review of "Clinical Results of the MINIject Implant for Suprachoroidal Drainage"

_jcm, 2024, doi:10.3390/jcm13102831_

Round 1

Reviewer 1 Report

Comments and Suggestions for Authors

Summary:

This retrospective study evaluated the safety and efficacy of the MINIject implant for suprachoroidal drainage in patients with insufficiently controlled glaucoma. The study aimed to assess its impact on intraocular pressure (IOP) reduction, changes in IOP medication, need for other glaucoma surgery, and adverse events. The results showed a significant reduction in IOP and IOP medication following implantation, with a low rate of adverse events.

Novelty: The study presents novel findings regarding the use of the MINIject implant for suprachoroidal drainage in glaucoma management, which contributes to the field's knowledge base.

  1. Significance of Content: The research addresses a clinically significant issue of managing glaucoma in patients with insufficiently controlled IOP, offering a potential new approach with the MINIject implant. Quality of Presentation: The manuscript is well-presented, with clear descriptions of the study objectives, methods, results, and conclusions. Scientific Soundness: The research design and methods are appropriate for evaluating the safety and efficacy of the MINIject implant, and the results are supported by the data presented. Interest to the readers: The study's findings are likely to be of interest to ophthalmologists, especially those involved in glaucoma management, as it introduces a new approach with promising results. Overall Merit: The manuscript has overall merit due to its relevance, scientific soundness, and potential impact on clinical practice in managing glaucoma.

Recommendation:
Based on the review, I recommend accepting the manuscript in its present form. The study's findings are significant, and further larger prospective studies with longer follow-up can confirm and build upon these results.

Comments for Authors:
The authors have conducted a valuable study evaluating the MINIject implant's efficacy for suprachoroidal drainage in glaucoma patients. The manuscript is well-written and provides clear insights into the study's objectives and outcomes. However, considering the importance of long-term follow-up in glaucoma management, it would be beneficial to discuss potential plans for future studies with extended observation periods to assess the implant's sustained efficacy and safety over time. Additionally, providing more details on patient demographics, such as age distribution and glaucoma severity, could enhance the manuscript's comprehensiveness.

Ethical Concerns:
While there are no significant ethical concerns identified, it's essential to ensure patient confidentiality and ethical conduct throughout the study and manuscript preparation process.

Reviewer 2 Report

Comments and Suggestions for Authors

This is an interesting study that investigated the clinical results of the MINIject implant for suprachoroidal drainage. The data are well-presented and progress logically, but several clarifications are needed, and some other comments should be considered.

1.     Page 1, line 21. The number of glaucoma cases worldwide has been increased to 80 M, please revise.

2.     Page 1, line 28. It’s a little bit confusing for the introduction of the outflow pathway. There are two outflow pathways: conventional (trabecular meshwork and Schlemm’s canal) and unconventional pathway. By distinguishing between these two pathways, we can better comprehend the intricacies of the outflow process. Please revise.

3.     Given that the study focused on eyes diagnosed with primary open-angle and pseudoexfoliative glaucoma, it would be beneficial to provide an overview of these two types of glaucoma.

4.     Could the authors please provide more details about the statistical analysis method that was utilized in this study?

5.     In Figure 2, should the label of the y-axis be probability (100%)
